# Weakly Supervised Volumetric Segmentation via Self-taught Shape Denoising Model

**Qian He**[*1,2,3]                                              HEQIAN@SHANGHAITECH.EDU.CN
**Shuailin Li**[*1]                                              LISHL@SHANGHAITECH.EDU.CN
**Xuming He**[1,4]                                              HEXM@SHANGHAITECH.EDU.CN

[1] *School of Information Science and Technology, ShanghaiTech University*

[2] *Shanghai Institute of Microsystem and Information Technology, Chinese Academy of Sciences*

[3] *University of Chinese Academy of Sciences*

[4] *Shanghai Engineering Research Center of Intelligent Vision and Imaging*

## Abstract

Weakly supervised segmentation is an important problem in medical image analysis due to the high cost of pixelwise annotation. Prior methods, while often focusing on weak labels of 2D images, exploit few structural cues of volumetric medical images. To address this, we propose a novel weakly-supervised segmentation strategy capable of better capturing 3D shape prior in both model prediction and learning. Our main idea is to extract a self-taught shape representation by leveraging weak labels, and then integrate this representation into segmentation prediction for shape refinement. To this end, we design a deep network consisting of a segmentation module and a shape denoising module, which are trained by an iterative learning strategy. Moreover, we introduce a weak annotation scheme with a hybrid label design for volumetric images, which improves model learning without increasing the overall annotation cost. The empirical experiments show that our approach outperforms existing SOTA strategies on three organ segmentation benchmarks with distinctive shape properties. Notably, we can achieve strong performance with even 10% labeled slices, which is significantly superior to other methods. Our code is available at: https://github.com/Seolen/weak_seg_via_shape_model.

**Keywords:** Weakly supervised segmentation, 3D shape prior, self-taught learning.

## 1. Introduction

Volumetric image segmentation is of great importance in many computer-aided medical applications, including auxiliary diagnosis and follow-up treatment. Recently, deep learning-based approaches (Milletari et al., 2016; Isensee et al., 2020) have achieved remarkable performance on semantic object segmentation in 3D medical images. However, the success of those supervised methods often requires a large quantity of images with pixelwise annotations, which are expensive and time-consuming to collect. In order to mitigate this problem, weakly-supervised methods have been explored (Rajchl et al., 2016; Kervadec et al., 2019, 2020), which typically convert the volumetric segmentation to a series of 2D segmentation tasks and use box or scribble annotation only to train a segmentation network for the 2D tasks.

Despite their promising results, those 2D-based approaches suffer from several limitations when applied to volumetric images (e.g., CT or MRI). Firstly, they simply stack the generated 2D masks

---

[*] Contributed equally

together as the final output, and thus tend to produce inaccurate object shape (Kervadec et al., 2019, 2020). In addition, they ignore the continuity of volumetric data in 3D space and are unable to exploit label correlation between consecutive 2D slices. Due to these limitations, 2D methods tend to perform worse than their 3D counter parts given sufficient volumetric data with small inter-slices spacing (Baumgartner et al., 2017). Furthermore, most weak annotations have a restrictive form (Bearman et al., 2016; Lin et al., 2016; Dai et al., 2015; Pinheiro and Collobert, 2015) and provide a poor guidance for learning object shape prior in medical images. More detailed discussions on related works are presented in Appendix A.

In this paper, we propose a novel weakly supervised learning strategy for volumetric object segmentation to tackle the aforementioned limitations. Our main idea consists of two aspects: First, we propose a self-taught learning method to capture the 3D shape prior of a target object class based on object mask augmentation. We then incorporate this learned shape prior into the training process of a shape-aware segmentation network. Second, we adopt a sparse annotation scheme to better exploit the spatial continuity of object mask and facilitate learning the shape context without increasing overall labeling cost.

To achieve this, we design a deep neural network consisting of two main modules: a Semantic Segmentation Network (SSN), which produces an initial 3D segmentation mask from the input image, and a Shape Denoising Network (SDN), which then refines the initial mask and outputs a final volumetric segmentation. To train the deep network, we first introduce a sparse weak annotation scheme, in which we annotate a specific subset of 2D image slices and design a hybrid label that integrates a foreground scribble and a loose bounding box of the target object. Given the weak labels, we then develop an iterative learning framework for our network model that alternates between pixelwise label generation and network parameter update.

Specifically, we first initialize our network model by training the segmentation module (i.e., SSN) using the weak labels, which generates initial segmentation masks for the training data. We then utilize the initial masks to learn the shape refinement module (i.e., SDN) with a self-taught method. To that end, we choose a mask prediction with the highest confidence as the target shape, and train the SDN module as a denoising autoencoder for the object masks (Vincent et al., 2010; Sundermeyer et al., 2018; Oktay et al., 2017). To generate the noisy mask input, we apply several noise augmentation schemes to the target shape based on empirical error patterns in the initial mask predictions. After the model initialization, our learning procedure performs a two-step update iteratively, including pixel-wise pseudo label generation followed by network learning. For label generation, we fuse the predictions of our SSN and SDN with an uncertainty filtering mechanism, which allows us to utilize the learned shape prior to improve label quality. In the network learning, we freeze the SDN and update the SSN with the supervision of both weak and the generated label.

We evaluate our method on three benchmarks with organs of distinctive shape properties: trachea in SegTHOR Challenge (Trullo et al., 2019), left atrium in 2018 Atrial Segmentation Challenge and prostate in Promise12 Challenge (Litjens et al., 2014). The empirical results show that our method consistently outperforms previous approaches. Moreover, we achieve strong results with a small amount of annotation (10% slices), when other existing methods would fail in that setting.

## 2. Method

We now introduce our method for weakly supervised volumetric segmentation. We start from the problem setting and model overview in Sec. 2.1, followed by the model design in Sec. 2.2. To

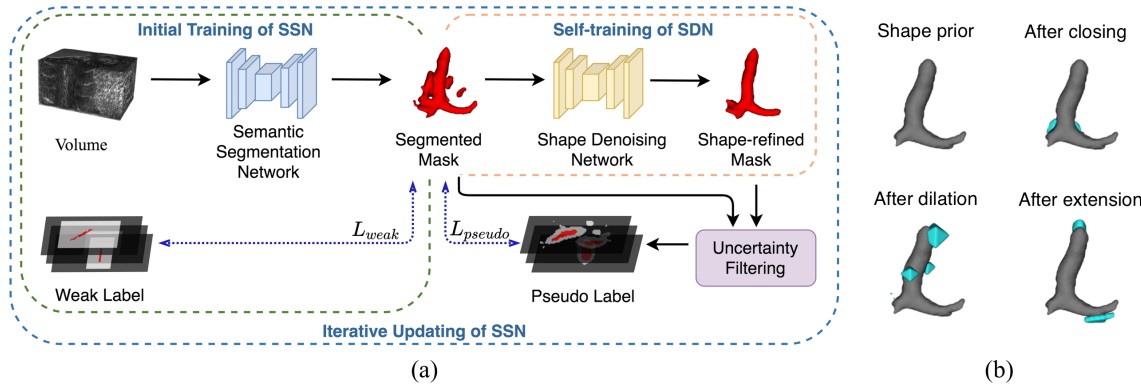

Figure 1: (a) Overview of our method. (b) An example of our self-taught shape representation and corresponding augmentation effect on trachea.

learn our network, we propose a weak annotation strategy in Sec. 2.3 and adopt an iterative learning framework in Sec. 2.4.

### 2.1. Problem Setting and Model Overview

Given an input volumetric image $\mathbf{I} \in \mathbb{R}^{H \times W \times D}$, our goal is to estimate its segmentation mask $\mathbf{M} \in \mathcal{S}^{H \times W \times D}$, where $H$ and $W$ are the height and width of an image slice, and $D$ is the number of slices. $\mathcal{S} = \{0, 1\}$ is the semantic label set with 0 as background and 1 as foreground. Assume we are given a training set $\mathcal{D} = \{\mathbf{I}^n, \mathbf{Y}^n\}_{n=1}^{N}$, where $\mathbf{Y}^n \in \mathcal{S}'^{H \times W \times D}$ is the corresponding weak label to $\mathbf{I}^n$, $\mathcal{S}' = \mathcal{S} \cup \{u\}$ with $u$ representing unlabeled pixels, and $N$ is the number of training data samples. We focus on single foreground class segmentation in this paper, while our method can be applied to multi-class problems by dealing with each class separately.

The main idea is to learn a self-taught shape prior from weak labels and then to utilize this prior for further shape denoising and refinement. To achieve this, we develop a deep neural network consisting of two main modules: a Semantic Segmentation Network (SSN) and a Shape Denoising Network (SDN). Our SSN first predicts an initial coarse segmentation mask from the input image volume and then our SDN applies the self-taught shape prior on this initial mask for denoising and refinement. To incorporate learned shape prior to further improve our model, we adopt an iterative learning framework, by generating pseudo labels and updating the model iteratively. An overview of our method is in Fig. 1.

### 2.2. Model Design

We now describe the two main network modules of our model in detail as below:

**Semantic Segmentation Network**   Our Semantic Segmentation Network (SSN) $\mathcal{F}_{SSN}$ provides an initial coarse mask by taking a volumetric image $\mathbf{I}$ as input and outputting a probability map $\mathbf{P}_s \in [0, 1]^{H \times W \times D}$, indicating the confidence of each pixel belonging to foreground. From $\mathbf{P}_s$ we can derive the initial foreground segmentation mask $\mathbf{M}_s$: $\mathbf{P}_s = \mathcal{F}_{SSN}(\mathbf{I}; \Theta), \mathbf{M}_s = \mathbb{1}(\mathbf{P}_s > 0.5)$, where $\Theta$ denotes the parameters of $\mathcal{F}_{SSN}$ and $\mathbb{1}(\cdot)$ is the indicator function. We instantiate our SSN with nnU-Net (Isensee et al., 2020), which is the state-of-the-art model architecture for medical image semantic segmentation. Detailed network configurations are in Appendix C.

Figure 2: We show different annotations for trachea (left) and left atrium (right). *Scribble (dilation)* denotes generated scribble by foreground mask dilation.

**Shape Denoising Network**   We design a Shape Denoising Network (SDN) $\mathcal{F}_{SDN}$ to encode a unified shape prior and then to apply to the initial coarse mask for shape refinement, inspired by Denoising Autoencoder (DAE) (Vincent et al., 2010) and Augmented Autoencoder (AAE) (Sundermeyer et al., 2018). Given the initial mask $\mathbf{M}_s$ from SSN output, our SDN implicitly applies self-taught shape prior constraints and outputs a clean and shape-refined mask $\mathbf{M}_d$: $\mathbf{P}_d = \mathcal{F}_{SDN}(\mathbf{M}_s; \Omega), \mathbf{M}_d = \mathbb{1}(\mathbf{P}_d > 0.5)$, where $\Omega$ denotes the parameters of $\mathcal{F}_{SDN}$. Since we aim for the final mask rather than the latent embedding, our $\mathcal{F}_{SDN}$ shares the same U-Net architecture as $\mathcal{F}_{SSN}$, which keeps a larger spatial resolution at its bottleneck and includes skip connections to capture more mask details.

### 2.3. Weak Annotation Strategy

In order to better exploit the spatial continuity of object mask and facilitate learning the shape context, we now introduce a sparse weak annotation strategy for the task of volumetric segmentation. Our annotation scheme consists of two components, including **slice selection** and a **hybrid labeling strategy**. For **slice selection**, we choose to label the starting and the ending slices of each foreground object, which include important boundary information in z-axis. Except for those two slices, we also randomly label a subset of slices in between. In this work, we investigate multiple labeling strategies with 10%, 30%, 50%, and 100% labeled foreground slices. Moreover, we design a **hybrid labeling strategy** for 2D slices, including a long axis scribble for the foreground object and a loose bounding box to encircle all foreground pixels in this slice. Specifically, for the long axis scribble, the annotators only have to click two points near the boundary from inside the foreground object, which can be automatically connected to a line. For the loose bounding box, the annotators only need to click two points from upper left corner to lower right corner, which can also be automatically connected into a box. Note that our hybrid label does not require precise localization of boundary points, but only needs the annotators to roughly point out the inside and outside regions of foreground. Compared to traditional scribble or tight bounding box, our hybrid label provides rich background information and rough localization, as well as foreground pixel label, with only four points for each 2D slice. To simulate our weak annotation strategy, we derive our annotations from full masks. Examples are shown in Fig. 2. More details are presented in Appendix B.

### 2.4. Model Learning

To effectively train our model, we adopt an iterative learning framework, by generating pseudo labels and updating the model iteratively. To generate initial pseudo labels, we first initialize our SSN and SDN. Then we compute pseudo labels with an uncertainty filtering mechanism on the combination of the outputs of our SSN and SDN, to remove noise and to incorporate learned shape

prior for model updating. Below we sequentially introduce the training of our SSN and our self-taught SDN, our uncertainty filtering for pseudo label generation, and finally our model updating.

**Training Semantic Segmentation Network** We first train our SSN on weak labels to provide initial segmentation masks, which serve as important training signals for our SDN. Given input images and corresponding weak labels, we train our SSN with weighted cross entropy on labeled pixels: $\mathcal{L}_{SSN}(\Theta) = \mathcal{L}_{wce}(\mathbf{P}_s, \mathbf{Y})$. Due to highly imbalanced foreground and background in weak labels, we adopt an auto-weighting strategy in our loss function. Specifically, for each volume with $N_b$ labeled background pixels and $N_f$ labeled foreground pixels, we compute the loss as $(\frac{1}{N_b} \sum_i^{N_b} l_i + \frac{1}{N_f} \sum_j^{N_f} l_j)/2$, where $l_i$ denotes cross-entropy loss of pixel $i$.

**Self-training of Shape Denoising Network** We train our SDN with a self-taught learning strategy, different from previous methods for denoising model learning. DAE (Vincent et al., 2010) applies artificial random noise to input images and reconstructs corresponding clean targets. AAE (Sundermeyer et al., 2018) proposed a domain randomization technique to mimic environment and sensor variations captured by real cameras. They utilize this technique to augment their input synthetic images and reconstruct images invariant to irrelevant factors other than orientation.

In our case, one main difference is that we have no full mask annotations for the self-taught learning. One possible solution is to use digital synthetic shape models, which might have a large domain gap to different datasets, especially for masks in medical segmentation. To avoid such domain gap, we propose a self-taught learning strategy, to first extract a self-taught shape representation with our weakly supervised segmentation model SSN and then train our SDN with this shape representation. The underlying assumption is that our trained SSN is able to generate masks with above-average accuracy for some instances in training split, and thus can supply those masks with better shape quality to help refine other masks. To this end, we first compute the average foreground probability of each $\mathbf{P}_s$ in training split as the confidence of each predicted mask $\mathbf{M}_s$, and then take the mask with the highest confidence as our self-taught shape representation $\mathbf{M}^*$ to train our SDN.

To train our SDN for noise removal, instead of adding general random noise to input masks, we specifically design our noise augmentation, based on the observation of errors produced by our initially trained SSN. We summarize the typical error modes of initial segmentation masks in three categories: (1) Over-smoothed regions; (2) Wrongly attached blobs; (3) Over-prediction of foreground regions beyond starting and ending slices. These errors occur mainly because there is no clear boundary supervision in weak labels, while the neighboring objects might share similar intensity or texture to the target object.

To equip our SDN with capability to deal with the aforementioned errors, we design three corresponding noise augmentation operations: (1) Closing; (2) Dilation; (3) Extension of marginal slices. Examples are as shown in Fig. 1 (b). Moreover, we apply spatial transformation including rotation, translation, and scaling, to capture rich position and size variations, which help learn the underlying shape manifold. Detailed information is in Sec. 3.2. We augment our self-taught shape representation $\mathbf{M}^*$ into $\hat{\mathbf{M}}^*$, and train our SDN to reconstruct the clean mask $\mathbf{M}^*$ with cross entropy loss: $\mathcal{L}_{SDN}(\Omega) = \mathcal{L}_{ce}(\mathcal{F}_{SDN}(\hat{\mathbf{M}}^*; \Omega), \mathbf{M}^*)$.

**Uncertainty Filtering** To incorporate the self-taught shape prior to further improve our model and to remove noise, we generate pseudo masks from predictions of both SSN and SDN with a simple uncertainty filtering mechanism. Specifically, we first compute the intersection of the segmented mask from SSN and the shape-refined mask from SDN, and then apply uncertainty filtering, based

on the confidence of each pixel in $\mathbf{P}_s$ from the semantic segmentation output. We compute pseudo label masks for foreground ($\mathbf{Y}_{fg}$) and background ($\mathbf{Y}_{bg}$) independently as below, where $\sigma_{fg}$ and $\sigma_{bg}$ are the respective uncertainty threshold and further explained in Appendix D. The final pseudo label $\mathbf{Y}_p$ combines $\mathbf{Y}_{fg}$ and $\mathbf{Y}_{bg}$, with unlabeled pixels set to $u$.

$$\mathbf{Y}_{fg} = \mathbf{M}_s * \mathbf{M}_d * \mathbb{1}(\mathbf{P}_s > \sigma_{fg}), \quad \mathbf{Y}_{bg} = (1 - \mathbf{M}_s) * (1 - \mathbf{M}_d) * \mathbb{1}(\mathbf{P}_s < \sigma_{bg}) \tag{1}$$

$$\mathbf{Y}_p = \mathbb{1}(\mathbf{Y}_{fg} = 1) + u * \mathbb{1}(\mathbf{Y}_{fg} = 0) * \mathbb{1}(\mathbf{Y}_{bg} = 0) \tag{2}$$

**Model Updating**  Given the generated pseudo label $\mathbf{Y}_p$, we update the parameters $\Theta$ of our SSN by minimizing the weighted cross entropy loss of segmentation probability $\mathbf{P}_s$ w.r.t. both the original weak labels and the generated pseudo labels: $\mathcal{L}(\Theta) = \lambda_w \mathcal{L}_{wce}(\mathbf{P}_s, \mathbf{Y}) + \lambda_p \mathcal{L}_{wce}(\mathbf{P}_s, \mathbf{Y}_p)$, where $\lambda_w$ and $\lambda_p$ are the corresponding loss weights. Note that we fix the parameters $\Omega$ of our SDN in iterative learning. Based on empirical observation, updating $\Omega$ does not provide further improvement. This is mainly because our self-taught shape representation is of relatively good quality and our noise augmentation is strong enough to capture various error modes.

## 3. Experiment

We evaluate our method on three public benchmarks with organs of distinctive shape properties, including trachea in SegTHOR (Trullo et al., 2019), left atrium in 2018 Atrial Segmentation Challenge, and prostate in PROMISE12 (Litjens et al., 2014). On each dataset, we compare with the state-of-the-art methods utilizing different weak annotations. Due to the page limit, we present results on PROMISE12 in Appendix E.

Below we first introduce dataset information in Sec. 3.1 and implementation details in Sec. 3.2. Then we present our experimental results comparing to other methods in Sec. 3.3, followed by comprehensive ablation study in Sec. 3.4. Moreover, we conduct further analysis on our SDN for better understanding of the shape denoising mechanism in Appendix F and discuss our failure cases and potential future work in Appendix G.3.

### 3.1. Datasets

**SegTHOR Challenge**  SegTHOR challenge[1] (Trullo et al., 2019) consists of 60 thoracic CT scans, from patients diagnosed with lung cancer or Hodgkin's lymphoma. It is an isotropic dataset with all scanner images in size of $512 \times 512 \times (150-284)$. The in-plane spacing varies from 0.90 mm to 1.37 mm and the z-spacing changes from 2 mm to 3.7 mm. We split 40 publicly available scans into 30 for training and 10 for validation, and evaluate on 20 testing scans using the official challenge page. There are four organs in this dataset: heart, aorta, trachea and esophagus. We conduct experiments on trachea for its challenging and representative organ shape.

**Left Atrium Dataset**  Left Atrium (LA) dataset is from 2018 Atrial Segmentation Challenge[2]. It contains 100 pairs of 3D gadolinium-enhanced MR imaging scans (GE-MRIs) and LA segmentation masks. All scans are isotropic and have spacing of $0.625 \times 0.625 \times 0.625 mm^3$ . The in-plane resolution of the MRIs varies for each patient, while all MRIs have exactly 88 slices in z-axis. We split 100 scans into 60 for training, 20 for validation, and 20 for testing.

For all datasets, we use the standard evaluation metric Dice coefficient (Dice).

---

1. https://competitions.codalab.org/competitions/21145
2. http://atriaseg2018.cardiacatlas.org/

Table 1: Quantitative results on the test splits of trachea and left atrium. All presented numbers are in Dice [%]. '−' under 10% denotes that BoxPrior failed in predicting any foreground.

| Method | Annotation | Trachea (Test) | | | | Left Atrium (Test) | | | |
|---|---|---|---|---|---|---|---|---|---|
| | | 100% | 50% | 30% | 10% | 100% | 50% | 30% | 10% |
| nnU-Net (Isensee et al., 2020) | Full label | 89.74 | | | | 92.63 | | | |
| BoxPrior(Kervadec et al., 2020) | Box | 79.82 | 48.78 | – | – | 83.93 | 83.51 | 81.86 | – |
| KernelCut(Tang et al., 2018) | Scribble* | 84.39 | 83.44 | 82.77 | 67.78 | 78.97 | 76.71 | 74.42 | 64.93 |
| Ours | Scribble* | 84.61 | 83.88 | 83.37 | 81.82 | 85.61 | 84.11 | 83.26 | 83.11 |
| KernelCut(Tang et al., 2018) | Hybrid | 84.74 | 83.55 | 83.38 | 76.43 | 77.54 | 76.72 | 73.64 | 67.27 |
| Ours | Hybrid | **85.54** | **83.97** | **83.78** | **83.19** | **86.31** | **86.25** | **83.81** | **83.41** |

## 3.2. Implementation Details

To fit 3D images into our network, we adopt a prior crop data preprocessing based on weak annotations. Specifically, we first resample image volumes into the same spacing, then align all training volumes based on their centers and pad them to the same size, and finally crop 1.2 times the size of the union cube of weakly annotated pixels from all volumes. All pixels outside our loose bounding boxes and slices beyond the starting and ending slices are used as background labels.

For our noise augmentation in training SDN, more detailed operations are as follows: (1) Closing. We use standard morphological closing by first dilating and then eroding images. Closing provides over-smoothed masks which is a common error for trachea and left atrium. (2) Dilation. We first randomly choose a center near mask boundary, and then dilate it with a random number of iterations. For anisotropic datasets like prostate, we adopt 2D dilation, while for isotropic datasets we utilize 3D dilation. (3) Extension of marginal slices. We elongate masks by simply copying a few starting or ending slices in z-axis. Random spatial transformations are used by default with a probability of 0.2 to enrich shape variations.

More detailed hyper-parameters and training settings are included in Appendix D.

## 3.3. Results

We compare our method using our proposed annotation, to state-of-the-art methods using scribble or box labels at the same cost. In Table. 1, we show quantitative results on labeled foreground slice ratios ranging from 100% to 10%. For fair comparison, different annotation types share the same labeled slices in each setting. Scribble* denotes using the same long axis of our hybrid label as foreground annotation and taking our loose box edges as background, while our hybrid also takes region out of boxes as background, encoding more shape context. We consider the cost of hybrid, scribble, and box for a slice roughly the same, and explain justification details in Appendix B.

Ours+Hybrid consistently outperforms KernelCut (Tang et al., 2018) and BoxPrior (Kervadec et al., 2020), especially on 10% labeled-slice setting, with a large gap of 15.41% on trachea and 18.48% on left atrium compared to KernelCut, where BoxPrior fails to predict any foreground due to extreme imbalance in their regularization loss terms. Note that our method is robust in different label density settings and our performance only decreases mildly with fewer labeled slices, which verifies the capability of our model to utilize feature correlation and self-taught shape prior to fill in missing labels. Moreover, KernelCut does not perform well on data without clear foreground boundaries and distinct neighborhood like left atrium, BoxPrior fails to handle small and distant multi-connected

Table 2: Ablation study on our model components. We conduct experiments in a drop-one-out manner.

| Method | Shape prior | Iterative | Trachea (Val) | | |
|---|---|---|---|---|---|
| | | | 50% | 30% | 10% |
| Ours | ✓ | ✓ | **83.45** | **83.18** | **83.18** |
| | CRF | ✓ | 81.08 | 80.48 | 80.36 |
| | – | ✓ | 78.97 | 78.59 | 78.11 |
| | ✓ | – | 74.91 | 74.80 | 74.17 |
| Baseline | – | – | 69.17 | 68.39 | 62.50 |

Table 3: Ablation study on annotations. We compare our hybrid label to different types of scribbles and box.

| Method | Annotation | Trachea (Val) | | |
|---|---|---|---|---|
| | | 50% | 30% | 10% |
| Ours | Hybrid | **83.45** | **83.18** | **83.18** |
| | Scribble* | 83.05 | 82.75 | 81.63 |
| | Scribble (20-50) | 78.91 | 75.80 | 75.45 |
| | Scribble (dilation) | 78.86 | 77.99 | 76.60 |
| | Box | 82.25 | 81.70 | 80.60 |

regions like trachea, while our method is robust to all these challenges. We also report the results of Ours+Scribble* and KernelCut+Hybrid, showing that our method can still outperform KernelCut with the same label and that shape context information encoded by hybrid label can further boost performance. Some qualitative results are in Appendix G.1.

### 3.4. Ablation Study

We conduct comprehensive ablation study on our model in Table. 2 and on hybrid label in Table. 3. Ablation on loss terms is in Appendix G.2.

**Shape prior** SDN utilizes a self-taught shape prior for shape refinement. CRF denotes replacing SDN with DenseCRF, which makes performance drop 2%-3%. Besides, DenseCRF post-processing on a 3D volume takes about 3.50s, while inference of our SDN only needs 0.02s. Moreover, removing SDN, i.e., removing $M_d$ and $(1 - M_d)$ in Eq. 1, results in a performance drop of 4%-5%.

**Iterative** Iterative learning incorporates learned shape prior to iteratively improve our segmentation model. Without iterative learning for refinement, model performance drops about 9%.

**Annotation** We compare our hybrid label to different types of scribbles and box. The results on hybrid label outperform other labels with considerable gaps, showing that with more shape context, hybrid is more informative than scribble or box at the same cost. All scribbles share the same foreground label as our hybrid label. Scribble* denotes taking our loose box edges as background, Scribble (20-50) also denotes adopting loose box edges but with larger distance of 20-50 pixels to its tight box, and Scribble (dilation) represents scribbles generated by ground truth foreground dilation. See Appendix B for more details. Scribble*, which directly derives from our hybrid label, achieves the best results compared to other scribble variations, showing that it is probably the most informative version of scribble at the same labeling cost. For Box, we first generate foreground and background labels from labeled slices with GrabCut, and use them as weak labels in our method.

### 4. Conclusion

In this work, we have developed a novel approach and proposed a sparse annotation scheme for weakly supervised volumetric segmentation. Our proposed self-taught shape denoising network is able to refine the segmented masks into better shapes and we improve our model performance by iteratively updating the model with learned shape prior. Moreover, our sparse annotation scheme provides more information comparing to other weak labels at the same labeling cost. Evaluation on three benchmarks with distinctive shape properties shows that our method robustly outperforms other state-of-the-art methods utilizing different weak annotations.

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

## Appendix A. Related Work

**Weakly Supervised Semantic Segmentation**    To reduce labeling cost, weakly supervised semantic segmentation (WSSS) uses coarse annotations, e.g., image-level labels (Wang et al., 2020; Fan et al., 2020; Chang et al., 2020; Sun et al., 2020; Kolesnikov and Lampert, 2016; Huang et al., 2018; Wang et al., 2018), bounding boxes (Khoreva et al., 2017; Song et al., 2019; Papandreou et al., 2015; Kervadec et al., 2020) and scribbles (Vernaza and Chandraker, 2017; Tang et al., 2018).

Existing methods can be roughly divided into two groups. The first group adopts an iterative learning framework (Papandreou et al., 2015; Rajchl et al., 2016; Cai et al., 2018; Can et al., 2018; Ji et al., 2019), which (iteratively) generate pseudo labels as supervision for their models. These approaches can (iteratively) obtain training signals from pseudo labels, but yet still suffer from error propagation. The second group avoids error propagation by applying a regularization-based framework (Pathak et al., 2015; Kervadec et al., 2020; Tang et al., 2018). BoxPrior (Kervadec et al., 2020) proposes several global constraints derived from box annotations to optimize the segmentation model. KernelCut (Tang et al., 2018) proposes to use MRF/CRF regularization loss terms for better mask predictions. (Peng et al., 2020) adopts a 3D segmentation framework with discrete constraints and regularization priors, but it requires an accurate global anatomical atlas which is hard to obtain in real scenario. Compared to iterative learning methods, these approaches are light-weighted but do not utilize learned information for further improvement of their models.

We adopt an iterative learning framework for its advantage of incrementally including training signals, and design a shape denoising model to mitigate the problem of error propagation. In addition, existing methods ignore intrinsic shape priors of objects, while we exploit a self-taught shape prior from weak labels to improve segmentation performance.

**Self-taught Learning**    Lack of training data is a common challenge in learning problems and self-taught learning is a promising solution. Recent works apply self-taught learning in classification (Raina et al., 2007; Wang et al., 2013; Feng et al., 2020), clustering (Li et al., 2017; Dai et al., 2008) and detection (Bazzani et al., 2016; Jie et al., 2017). We introduce self-taught learning to weakly supervised segmentation. We first extract a self-taught shape representation by leveraging weak labels with a segmentation network and then utilize a shape denoising network to encode this representation for further shape denoising and refinement.

**Denoising Autoencoder**    Denoising Autoencoder (DAE) (Vincent et al., 2010) encodes an image into a latent embedding which is invariant to noise, to represent the original clean image. Augmented Autoencoder (AAE) (Sundermeyer et al., 2018) produces the orientation encoding of the object in the input image, which is invariant to other transformation and environmental conditions. Different from these methods aiming for a representative embedding, our goal is to recover a clean and complete shape from an input mask. Our method implicitly captures the underlying manifold of true shape data, instead of images or object orientations. ACNNs (Oktay et al., 2017) also investigates modeling shape prior with an autoencoder for fully supervised segmentation and image super resolution, as regularization constraints in encoding space, while we develop a self-taught learning method and design a shape denoising autoencoder to explicitly perform denoising and to recover the clean shape.

Table 4: Statistics of 2D foreground mask size in three datasets, presented by calculating the length of all tight bounding box edges in pixel.

|  | Trachea | Left Atrium | Prostate |
|---|---|---|---|
| mean ($\pm$std) | 24 ($\pm$14) | 82 ($\pm$41) | 86 ($\pm$42) |

Table 5: Network configurations for trachea, left atrium and prostate datasets.

|  | Trachea | Left Atrium | Prostate |
|---|---|---|---|
| Input spacing (mm) | $2.50 \times 1.63 \times 1.63$ | $0.63 \times 1.33 \times 1.33$ | $2.20 \times 0.72 \times 0.72$ |
| Input resolution | $128 \times 112 \times 128$ | $96 \times 160 \times 128$ | $40 \times 160 \times 192$ |
| Pooling strides | $[[2,2,2],[2,2,2],[2,2,2],[2,2,2],[2,1,1]]$ | $[[2,2,2],[2,2,2],[2,2,2],[2,2,2]]$ | $[[1,2,2],[1,2,2],[2,2,2],[2,2,2],[2,2,2]]$ |
| Convolution kernel sizes | $[[3,3,3],[3,3,3],[3,3,3],[3,3,3],[3,3,3],[3,1,1]]$ | $[[3,3,3],[3,3,3],[3,3,3],[3,3,3],[3,3,3]]$ | $[[1,3,3],[1,3,3],[3,3,3],[3,3,3],[3,3,3],[3,3,3]]$ |

## Appendix B. Weak Annotation Strategy

To simulate our weak annotation strategy, we derive our annotations from full masks. For **foreground long axis**, we generate six lines across the mass center of a mask, to uniformly split the mask with intersection angles of 30 degree, and select the longest line within the mask, leaving distance of 5 pixels from line ending points to mask boundary. For slices with multiple connected foreground components, we only randomly label one. For our **loose bounding box**, we generate each edge of the box with distance of 10-20 pixels to the corresponding tight bounding box edge. The statistics in Table. 4 show that a threshold of 10-20 pixels for loose box is reasonable in practice. We also generate scribble and tight box for comparison. Note that our hybrid label can also be used as scribble. Another generated scribble shares the same foreground label as our long axis, but uses a curve denoting background label from dilation of the foreground mask with 20-50 iterations. All scribbles are in width of three pixels.

We consider the annotation cost of hybrid, scribble, and tight bounding box for a slice roughly the same, and explain details as below. For our hybrid label, we require four points to label a slice as described in Sec. 3.2. According to (Bearman et al., 2016), it takes workers a median of 2.4s to click on the first instance of an object, and 0.9s to click on every additional instance of an object class in PASCAL VOC 2012 (Everingham et al., 2010), which makes our hybrid label roughly $2.4 + 0.9 \times 3 = 5.1$s. For scribble, (Bearman et al., 2016) also claims that for every present class, it takes 10.9s to draw a free-form scribble on the target class. In Table. 1 and Table. 6, Scribble* denotes using the same long axis of our hybrid label as foreground annotation, while taking our loose box edges as background. This makes the cost of Scribble* the same as our hybrid label. Moreover, we conduct further experiments on different scribbles in Sec. 3.4, showing that Scribble* is the most effective and informative type of scribble comparing to some other options. For tight bounding box, (Papadopoulos et al., 2017) proposed extreme clicking for box annotation, which takes 7s to click on four extreme points to annotate a box. Based on the evidence above, we claim our hybrid label, scribble, and tight bounding box share roughly the same labeling cost.

## Appendix C. Network Architecture

For our SSN, we utilize a 3D U-Net structure following nnU-Net (Isensee et al., 2020), including an encoder and a decoder. Our detailed network architectures for each dataset are shown in Fig. 3 and the corresponding network configurations are shown in Table. 5. Both the encoder and the decoder

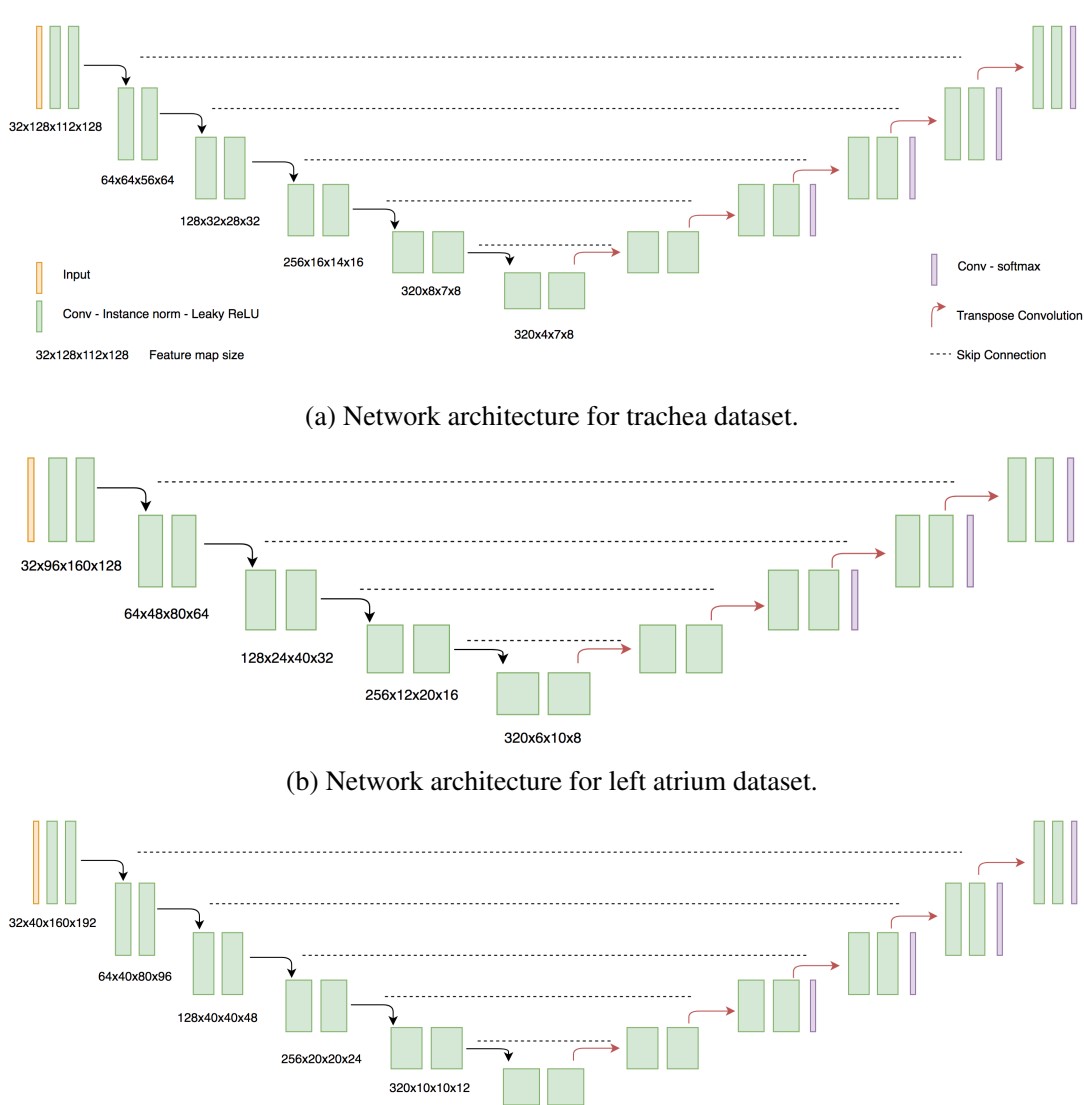

(a) Network architecture for trachea dataset.

(b) Network architecture for left atrium dataset.

(c) Network architecture for prostate dataset.

Figure 3: Network architectures. We adopt the same architecture for Semantic Segmentation Network (SSN) and Shape Denoising Network (SDN) but with independent parameters.

consist of four or five layers, depending on the input resolution. We define a computation block with three operations in sequence: conv - instance norm - leaky ReLU. Each layer contains two computation blocks which do not change feature spatial resolution. We implement downsampling with strided convolutions and upsampling with transposed convolutions. For our SDN, we adopt the same network architecture as our SSN with independent parameters.

---

**Algorithm 1** Training procedure.

---

**Input**: Weakly-labeled training set $\mathcal{D} = \{\mathbf{I}^n, \mathbf{Y}^n\}_{n=1}^{N}$;
**Output**: Network parameters $\Theta, \Omega$

  /* Initialize SSN */
  Train SSN with weak labels;
  /* Train SDN */
  Compute confidence of all predicted masks by SSN on training split;
  Select the mask with the highest confidence as the self-taught shape representation;
  Augment the shape representation with designed noise and spatial transformation as input;
  Train SDN to reconstruct the clean shape;
  /* Iterative learning */
  Repeat:
      Generate pseudo labels by combining outputs of SSN and SDN with uncertainty filtering;
      Update SSN with generated pseudo labels and weak labels;
  Until reaching the maximum epoch

---

## Appendix D. Model Training

We present the training procedure in Algorithm.1. Following nnU-Net (Isensee et al., 2020), we utilize the same image augmentation and deep supervision for model training, with batch size of 2. To train our model, we use the SGD optimizer. For initialization, we train SSN with initial learning rate of 1e-2 and decay it to 1e-3 in "poly" learning policy, for 200 epochs. To train SDN, we use constant learning rate of 1e-2 for 100 epochs. In iterative learning, we use slightly different parameters for different datasets. In uncertainty filtering, for each volume, we first sort the pixels in the predicted segmentation confidence map $\mathbf{P}_s$, and then set the uncertainty threshold $\sigma_{fg}$ to filter out the less confident 70% of all predicted foreground pixels for trachea. The ratio for left atrium and prostate is 50%. The corresponding $\sigma_{bg}$ for the background of each dataset is set to filter out double pixels of filtered foreground. In model updating, for trachea, we set loss weights $(\lambda_w, \lambda_p)$ as (1, 100), and train our model for a maximum of another 300 epochs with learning rate of 1e-3. As for prostate, we set $(\lambda_w, \lambda_p)$ as (0.1, 10) and learning rate as 1e-2. For left atrium, we set $(\lambda_w, \lambda_p)$ as (0.1, 10) and learning rate as 1e-3.

## Appendix E. Experiment on PROMISE12

**PROMISE12 Challenge** PROMISE12 challenge (Litjens et al., 2014) contains 50 transversal T2-weighted MR images in multiple scanning protocols[3], with the segmentation target prostates in the central area of images. All cases in this dataset are anisotropic, with spacing ranging from $2 \times 0.27 \times 0.27 mm^3$ to $4 \times 0.75 \times 0.75 mm^3$. Following the same setting as (Kervadec et al., 2020), we split 50 scans into 40 for training and 10 for validation, and report results on the validation split as testing is no longer available.

    Quantitative results are shown in Table. 6. On an organ with relatively simple shape like prostate, our method also consistently outperforms KernelCut and BoxPrior, especially with a large margin on 10% labeled-slice setting.

---

3. https://promise12.grand-challenge.org

Table 6: Quantitative results on the validation split of PROMISE12. All presented numbers are in Dice [%]. '–' under 10% denotes that BoxPrior failed in predicting any foreground.

| Method | Annotation | Prostate (Val) | | | |
|---|---|---|---|---|---|
| | | 100% | 50% | 30% | 10% |
| nnU-Net (Isensee et al., 2020) | Full label | 91.11 | | | |
| BoxPrior(Kervadec et al., 2020) | Box | 83.82 | 80.60 | 76.93 | – |
| KernelCut(Tang et al., 2018) | Scribble* | 78.68 | 77.13 | 76.72 | 72.84 |
| Ours | Scribble* | 85.55 | 84.09 | 83.87 | 80.59 |
| KernelCut(Tang et al., 2018) | Hybrid | 80.18 | 77.90 | 77.58 | 73.45 |
| Ours | Hybrid | **86.01** | **85.71** | **85.56** | **80.89** |

To compare our method to Boxprior on Hybrid, we conducted experiments on the 100% setting of prostate dataset. We added a Partial Cross-Entropy (PCE) loss to Boxprior to train on Hybrid, as BoxPrior+PCE+Hybrid, and the result is 72.56%, which is much lower than Ours+Hybrid. Moreover, to investigate the contribution of the PCE loss and the original BoxPrior loss, we conducted experiments of PCE loss only and BoxPrior loss only on Hybrid. With PCE loss only, the result is 73.15% and already higher than BoxPrior+PCE+Hybrid. With BoxPrior loss only, the result is 59.82%. Note that we tried extensive hyper-parameters tuning for BoxPrior+PCE+Hybrid, including their $w$ parameter, weight of each loss term, and learning rate, based on their experiments on thick boxes in their code.

## Appendix F. Analysis on Shape Denoising Network

In this section, we further discuss our Shape Denoising Network (SDN). The design of our SDN is based on the assumption that our Semantic Segmentation Network (SSN) trained on weak labels is able to provide initial masks, which would serve as a good starting point for SDN. We empirically found that some of the instances in the training split "look like" they have better quality than other instances, i.e., with clean and complete shapes. Then we compute the confidence of each predicted mask by calculating the average probability over its foreground pixels and found that the ones with the highest confidence usually have the best quality in shape. Therefore, we take the mask with the highest confidence predicted by SSN as the correct shape to train our SDN. Our SDN is never updated thereafter, mainly because the initial selected mask has good enough quality in shape, and empirically we found that updating SDN with new shapes did not provide further improvement.

Note that we do not use any ground truth masks of training data in our model training or inference, but we can retrospectively verify if the selected mask is of good quality by calculating its dice with the ground truth. Take the 30% setting on Trachea for example, the dice of the selected shape is 87.89%. After the entire process of our model learning, its dice only improves to 90.62%. which is not much changed in its shape.

We further analyze our SDN design in Table. 7. All of our augmentation operations contribute to improving SDN for noise and error removal, especially with dilation which increases performance by 3.83%. Moreover, we utilize shape instances with different confidence from SSN predictions to train our SDN. Comparison among the results of using shapes with different confidence ranks shows that the most confident case provides the best shape prior information, while our design is also robust to the shape selection criteria. Choosing a shape prediction with relatively low confidence (case rank

Table 7: Analysis of our SDN on the validation split of trachea dataset with 30% labeled slices. We present the contribution of each augmentation operation in an incremental manner. Moreover, we also show the effect of using different cases as our shape representation for training. Case rank denotes the confidence rank of the selected shape representation.

| | Case rank | Closing | Dilation | Extension | Dice [%] |
|---|---|---|---|---|---|
| Baseline | 1 | – | – | – | 68.39 |
| | 1 | ✓ | – | – | 69.19 |
| | 1 | ✓ | ✓ | – | 73.02 |
| Our SDN | 1 | ✓ | ✓ | ✓ | **74.80** |
| | 2 | ✓ | ✓ | ✓ | 74.62 |
| | 15 | ✓ | ✓ | ✓ | 74.05 |
| | 30 | ✓ | ✓ | ✓ | 72.62 |
| | Top 1-3 | ✓ | ✓ | ✓ | 74.08 |
| | Top 1-5 | ✓ | ✓ | ✓ | 73.98 |

30) still provides improvement to baseline, which shows that our noise augmentation and denoising learning is still able to perform well without a high-quality shape representation.

From the observation of initially trained SSN, we found that organs within a dataset typically have very similar shapes. The main variations can be captured by some mild spatial transformations, including translation, rotation, and scale. Therefore, we augment a single selected shape with random spatial transformation to form the shape distribution. Empirically we found that using a single shape with augmentation is sufficient for this task. We conducted experiments where we selected top 1, 1-3, or 1-5 shapes with the highest confidence to train the SDN, and using more shapes does not provide further improvement. For volumetric segmentation with large variety in shape, our method can potentially extend to multi-shape learning, e.g., resorting to standard clustering methods to obtain several templates, training an independent autoencoder for each shape, and obtain final results by voting based on similarity.

## Appendix G. Further Analysis

### G.1. Qualitative Results

We visualize some qualitative results in Fig. 4, demonstrating that our method provides masks with cleaner shape compared to other 2D methods.

### G.2. Ablation on Loss Terms

We also investigate the effect of our two loss terms for model training. As shown in Table. 8, our method with only pseudo label for iterative refinement training can significantly outperform baseline by more than 10%. With both weak label and pseudo label, our method can further improve performance by 1%-2%.

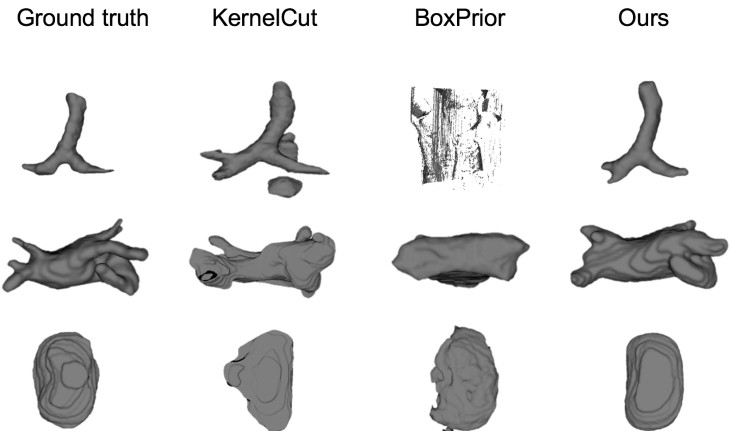

Figure 4: Qualitative results on three datasets with 30% labeled slices. **Top to bottom**: trachea, left atrium, and prostate.

Table 8: Ablation study on two loss terms of our method.

| Method | Weak label | Pseudo label | Trachea (Val) | | |
|---|---|---|---|---|---|
| | | | 50% | 30% | 10% |
| Baseline | ✓ | – | 69.17 | 68.39 | 62.50 |
| | – | ✓ | 81.89 | 81.57 | 81.14 |
| Ours | ✓ | ✓ | **83.45** | **83.18** | **83.18** |

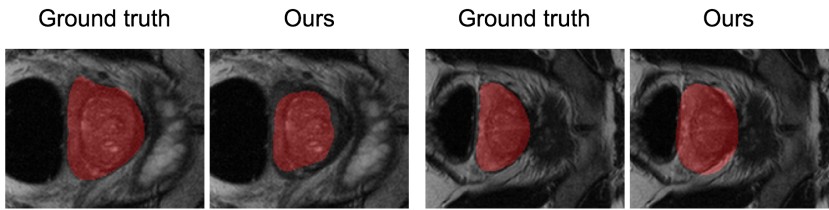

Figure 5: Two failure cases on prostate dataset.

## G.3. Failure Case and Future Work

We show two failure cases in Fig. 5, which have incomplete or over-predicted masks, despite the obvious intensity similarity or boundary in these slices. This is mainly because our model trained on weak labels, explores no boundary information or constraints during training. This could be further improved by incorporating low-level image feature such as boundary or superpixel into our model.

