# OpenReview forum: "Weakly Supervised Volumetric Segmentation via Self-taught Shape Denoising Model"
_MIDL.io/2021/Conference — MIDL 2021_

### Official Review · AnonReviewer3 · 2021-03-08

**Confidence:** 3
**Preliminary Rating:** 3
**Final Rating:** 3

**Summary:**

In this submission the authors propose a novel strategy for 1) weakly labelling 3D medical data (longitudinal scribble + bounding box), and 2) a training framework to learn from weak labels. The basic idea is to use a specialised denoising autoencoder (where the noise is morphological operations) to learn the desired shape of the structure on the fly, the final segmentation is then refined in a number of iterations using information from this AE. It is shown that the strategy performs very well in comparison to recent related work, especially when using few labelled input slices.

To me certain steps of the method seem heuristic and not well justified, however was thoroughly evaluated and appears to perform well in this setting.

**Strengths:**

Positive points:
 + Well written
 + Well motivated
 + Thorough evaluation on 3 datasets with recently published baselines and ablation study
 + Convincing results when using very few annotations
 + Thorough related work section (albeit in the Appendix)

**Weaknesses:**

Negative points:
 - Some points about the method remain unclear to me  (see detailed comments)
 - There is a lack of a theoretical (or intuitive) explanation why the proposed method works as it does
 - No guarantees of convergence to a stable solution

**Deanonymize Review:**

no

**Detailed Comments:**


Did I understand correctly that the "Uncertainty Filtering" and "Model Updating" steps get iterated? If so please clarify the following questions:
 - How many iterations were performed?
 - How was the model improvement over subsequent iterations (i.e. did it oscillate or converge, and if the latter then how quickly)?
 - How was the stopping criterium chosen?

I am having trouble following the intuition for why the proposed approach works as described. A couple of specific points:
 - Why does the Shape Denoising Network learn a useful shape model? To my understanding it is trained on the initial networks prediction which was only trained on the weak labels, and is never updated thereafter. Why should it know about the correct shape?
 - I am having difficulties understanding why the noise in form of morphological operations given to the Denoising Network leads it to produce cleaned-up shapes. It learns to undo opening/closing operations, but should have no concept of the true underlying shape. In my opinion, this is different from the reasoning in the original denoising autoencoder paper where the true data are corrupted with noise, forcing the DAE to learn the underlying manifold. Here we do not have the true data.
 - If I understood correctly, in Appendix A (related work), it is suggested that the proposed method is a type of EM, which would be nice because it would give a theoretical guarantee that the method converges. However, I am not sure, I see how it is EM given the heuristic nature of the shape denoising "prior".
These are my biggest issues with the submission in its current form and, in my opinion, giving a better intuition or even theoretical justification why we should expect this algorithm to work would make this a very strong submission.

I believe this sentence contains a key piece for the motivation of the approach: "The underlying assumption is that our trained SSN is able to generate masks with above-average accuracy for some instances in training split, and thus can supply those masks with better shape quality to help refine other masks". However, I do not understand what is meant. Could you please clarify?


**Final Rating Justification:**

The rebuttal has cleared some things up for me, and I am now am confident in my accept vote. The method is as previously stated relatively heuristic making it in my mind a good, but not very good contribution. Hence the "weak accept".

**Justification Of The Preliminary Rating:**

This is a very well written and well motivated paper, with a thorough evaluation, recent baselines and good results especially in the low-annotation regime. To me the method seems very heuristic and lacks theoretical guarantees, which is the main reason why I did not rate the submission "Strong accept". However, it is an interesting proof-of-concept and I believe will be of interest to the community.

**Paper Type:**

methodological development

**Special Issue:**

no

---

> ### Author Response · Authors · 2021-03-18
> **Reply to reviewer 3 (Part 1)**
>
> We thank reviewer 3 for the positive comments and the inspiring questions. The main concerns are about the intuitive or theoretical explanations for why the proposed approach works. Please find our detailed answers as below.
>
> **Detailed Comments:**
> > Did I understand correctly that the "Uncertainty Filtering" and "Model Updating" steps get iterated?
>
> Yes, the "Uncertainty Filtering" and "Model Updating" steps get iterated.
>
> > How many iterations were performed?
>
> For each setting, we perform a maximum of 300 epochs of iterative learning, where each epoch consists of 250 iterations of "Uncertainty Filtering" and "Model Updating" on a minibatch. We follow the epoch separation of nnU-Net [1] and decay the learning rate after each epoch.
>
> [1] Fabian Isensee, Paul F. Jaeger, Simon A. A. Kohl, J. Petersen, and Klaus Maier-Hein. nnu-net: a self-conﬁguring method for deep learning-based biomedical image segmentation. Nature methods, 2020.
>
> > How was the model improvement over subsequent iterations (i.e. did it oscillate or converge, and if the latter then how quickly)?
>
> Our model performance gradually improves and saturates after 50 epochs, and then reaches the peak with some mild oscillation. The oscillation is mostly within 5-15% in dice and is different for each dataset.
>
>
> > How was the stopping criterium chosen?
>
> For all settings of all datasets, we choose a unified maximum training epochs of 300, where the dice curves on validation splits reach plateau. After training, we utilize the validation split to choose the model checkpoint with the highest dice as our final model, and then apply it to the test split. The same stopping criterium is also adopted by other related works (KernelCut and Boxprior, in their code).
>
>
>
> > Why does the Shape Denoising Network learn a useful shape model? To my understanding it is trained on the initial networks prediction which was only trained on the weak labels, and is never updated thereafter. Why should it know about the correct shape?
>
> The design of our Shape Denoising Network (SDN) is based on the assumption that our Semantic Segmentation Network (SSN) trained on weak labels is able to provide initial masks, which would serve as a good starting point for SDN. We empirically found that some of the instances in the training split "look like" they have better quality than other instances, i.e., with clean and complete shapes. Then we compute the confidence of each predicted mask by calculating the average probability over its foreground pixels and found that the ones with the highest confidence usually have the best quality in shape. Therefore, we take the mask with the highest confidence predicted by SSN as the correct shape to train our SDN. Our SDN is never updated thereafter, mainly because the initial selected mask has good enough quality in shape, and empirically we found that updating SDN with new shapes did not provide further improvement.
>
> Note that we do not use any ground truth masks of training data in our model training or inference, but we can **retrospectively** verify if the selected mask is of good quality by calculating its dice with the ground truth. Take the 30% setting on Trachea for example, the dice of the selected shape is 87.89%. After the entire process of our model learning, its dice only improves to 90.62%. which is not much changed in its shape.

---

> ### Author Response · Authors · 2021-03-18
> **Reply to reviewer 3 (Part 2)**
>
>
> > I am having difficulties understanding why the noise in form of morphological operations given to the Denoising Network leads it to produce cleaned-up shapes. It learns to undo opening/closing operations, but should have no concept of the true underlying shape. In my opinion, this is different from the reasoning in the original denoising autoencoder paper where the true data are corrupted with noise, forcing the DAE to learn the underlying manifold. Here we do not have the true data.
>
> We believe our shape denoising network approximately captures the true underlying shape. As explained above, our selected shape is of good quality and more importantly, it is augmented with spatial transformation, which can cover a reasonable range of shape variations and form the manifold of the true shape data.
>
> Additionally, if the shape denoising network had no concept of the true underlying shape, it would not be able to distinguish a wrongly attached blob (from augmentation or SSN prediction) from a part of the true shape, and might randomly remove some parts of the input mask.
>
>
> > If I understood correctly, in Appendix A (related work), it is suggested that the proposed method is a type of EM, which would be nice because it would give a theoretical guarantee that the method converges. However, I am not sure, I see how it is EM given the heuristic nature of the shape denoising "prior". These are my biggest issues with the submission in its current form and, in my opinion, giving a better intuition or even theoretical justification why we should expect this algorithm to work would make this a very strong submission.
>
> We thank the reviewer for pointing out this and the "EM" in Appendix A (related work) is ambiguous. We meant to refer to these methods as "EM-like" and not all of them are proved to be EM. Our work takes an iterative learning framework similar to EM, but does not converge theoretically like EM. We will clarify the terminology in our revision.
>
> As for why we should expect this algorithm to work, we explain our intuition as follows:
> - First of all, with the initially trained SSN on weak labels, we can select a mask with high-quality shape based on the confidence of these initial masks from training split.
> - Then we train our SDN on the selected mask with noise augmentation, to encode the shape prior and to capture the underlying shape manifold. After training, SDN is able to recover masks in better shape given noisy inputs. More specifically, the initially predicted masks of SSN with worse shape quality than the selected mask can be improved by the trained SDN.
> - Then we incorporate the self-taught shape prior into an iterative learning framework and improve the SSN with shape information. Specifically, we utilize the shape-refined masks to help generate pseudo labels as new training signals for SSN.
> - To mitigate error propagation by pseudo labels, we adopt an uncertainty filtering mechanism.
> - In each iteration, segmented masks by SSN with lower shape quality are first refined by SDN into better shapes, and then feed back to improve SSN.
> - The shape quality of predicted masks by SSN gradually improves and the outputs of SDN become stable. Till then the pseudo labels also become stable and the model performance finally saturates.
>
>
> > I believe this sentence contains a key piece for the motivation of the approach: "The underlying assumption is that our trained SSN is able to generate masks with above-average accuracy for some instances in training split, and thus can supply those masks with better shape quality to help refine other masks". However, I do not understand what is meant. Could you please clarify?
>
> This is indeed the key motivation of our self-taught learning method. As explained above, we empirically found that we can select a mask with high quality in shape from predictions of initially trained SSN, with a simple selection scheme based on the foreground confidence of each predicted mask. Then we can encode the shape prior and capture the underlying shape manifold with our SDN and noise augmentation design. After that, the trained SDN is able to refine masks predicted by SSN that are originally in worse shapes. Finally, the refined masks are feed back in an iterative learning process to further improve SSN and segmentation results, till the model improvement finally saturates.

---

### Official Review · AnonReviewer1 · 2021-03-09

**Confidence:** 5
**Preliminary Rating:** 1
**Final Rating:** 3

**Summary:**

This paper is about weakly supervised segmentation, in binary 3D settings. The authors point out (rightfully) that a lot of existing methods happen to be simply "stacked" 2D segmentation, discarding potentially valuable 3D information while possibly causing inconsistent predictions between slices.

The authors propose a two-stage network, trainable end-to-end, composed of i) a standard nnU-Net for segmentation ii) a shape denoising network

The method is evaluated on three difference datasets, and compare to two existing methods (BoxPrior and KernelCut). However, due to issues in the experimental protocol, the results are difficult to interpret.

**Strengths:**

- The evaluation is performed on three different datasets
- The litterature is well presented and the authors list all the relevant works and their current limitations.
- The proposed method can perform well with only a fraction of the slices annotated
- Several ablation studies on their method (second stage, comparison to CRF, different kind of labels)

**Weaknesses:**

My main grip is the comparison between methods, that uses different types of labels for each, making it difficult to properly compare them. Currently, the comparison is as follow:
- Boxprior is trained with _tight_ bounding boxes
- KernelCut is trained with scribbles
- Theirs is trained with scribbles + _loose_ bounding boxes

I would argue that loose boxes + scribbles (providing info on 2 classes) give more information that tight boxes only (providing certainty on 1 class only). Both BoxPrior/DeepCut, and KernelCut [1], can readily benefit from the scribbles + loose boxes setting, and their parameters can readily be tuned to deal with the 'looseness' of the box [2] (which, at 10/20 pixels, isn't that loose). **Comparing all methods on the same labels would give much better insights.**

Moreover, drawing boxes in 3D should be included in the comparison, as it provides (potentially quite loose) box annotations on all slices, at only a fraction of the cost (even less than the "10% of annotated slices setting"). All methods can be evaluated on such labels.

Other, smaller issues (no specific order):
- The training procedure (for both networks) should be clearer. An algorithm in the appendix would help a lot.
- The paper also lacks a discussion on the computational cost of adding another, second stage network. Can you provide some element of comparison, both at training time and inference ?

---
[1] For the box prior, you can simply add a partial cross-entropy. For deep cut, add a partial cross-entropy and "correct" the full labels proposals with the scribbles; this also solve the vanishing proposals issue. For KernelCut, simply perform a partial cross-entropy on the outside of the box (background)

[2] For boxprior, its $w$ parameter can be used to deal with bigger gaps, as mentioned on their ablation study. DeepCut can take more time to tune (the 5 parameters of the CRF), but it is feasible (though annoying). KernelCut doesn't need any additional tuning in its hyper-parameters.

**Deanonymize Review:**

no

**Detailed Comments:**

I personally think that the main contribution of the paper is the second-stage network, that can be trained to perform a "clean-up" of the predicted segmentation. I would therefore to present it as a pluggable (but still end-to-end trainable) module that can go after any base network (be it fully or weakly supervised).

KernelCut, in that respect, is interesting with its CRFLoss; it can be both a base training method or a fine-tuning tool for another method. The updated comparison should reflect that.


I would also encourage the authors to release their code, for an easier use by the community.

The comments on performing 3D segmentation as a series of 2D segmentation are on point, but would benefit from some nuance. Depending on the inter-slices spacing, the correlation between slices can be low, and could hamper performances in 3D. This was actually noted in [1]; in some tasks 3D methods do not necessarily perform better than their 2D counterparts. 3D methods can also suffer from a lack of training data, due to the lower number of training samples. All of that vary between applications, and it does not invalidate what the authors claims, but having a bit more nuance would be beneficial.

---
[1] Baumgartner, C. F., Koch, L. M., Pollefeys, M., & Konukoglu, E. (2017, September). An exploration of 2D and 3D deep learning techniques for cardiac MR image segmentation. In International Workshop on Statistical Atlases and Computational Models of the Heart (pp. 111-119). Springer, Cham.

**Final Rating Justification:**

The authors did their best and worked hard, in only a week, to address my criticisms.

I still disagree with the authors on some points (notably the 3D boxes, which are trivial to get at the same cost, given their current annotation strategy--sec 2.1 "for slice selection we choose to label the starting and the ending slices of each foreground object"; among other things), but that will do for the conference version.

I cannot change that here, but **I am recommending this paper for the special issue**. This is where I believe several iterations could make an excellent paper.

For the camera ready version, **Table 2 and Algorithm 1 should be much more prominent, and go side by side.** This is what allows the reader to truly understand the sub-components of the method and their interaction.

**Justification Of The Preliminary Rating:**

The contribution of the authors are probably valuable, but the current experiments makes it difficult to assert how much. The contribution on the second stage network, compared to previous works, are also unclear.

A (major) revision of that paper could make an excellent re-submission somewhere, but there is too much modification and new results required for this round of reviews.

**Paper Type:**

both

**Questions To Address In The Rebuttal:**

In no particular order:
- The paper would benefit a lot from detailing more the _self-taught shape denoising network_, which is the main contribution of the paper. Could you develop that during the rebuttal, and highlight the differences with (Vincent et al., 2010) and (Sundermeyer et al., 2018) ?
- How would your method generalize to multi-class settings ?
- How would your method behave if it was trained with only tight boxes ?

**Special Issue:**

no

---

> ### Author Response · Authors · 2021-03-18
> **Reply to reviewer 1 (Part 1)**
>
> We thank reviewer 1 for recognizing the value of our contribution and the effectiveness of our method. The main concerns are about the comparisons of our work with some related works. Please find detailed answers to these concerns as below.
>
> **Weaknesses:**
> > Comparing all methods on the same labels (i.e., our proposed hybrid labels: loose boxes + scribbles).
>
> Most existing methods for weakly supervised segmentation are originally designed for the specific weak labels. In this paper, we focus on the task of weakly supervised volumetric segmentation, and fairly compare to other weakly supervised methods on different labels at the same labeling cost.
>
> However, as suggested by reviewer 1, we still applied KernelCut and BoxPrior on our hybrid labels and the results are in the tables below:
>
>
> A. On Trachea test set
>
> | Method | Annotation | 100%	| 50%	| 30%	| 10% |
> | :-----:| :----: | :----: | :----: | :----: | :----: |
> | KernelCut      | Hybrid    | 84.74  | 83.55  | 83.38  | 76.43  |
> | Ours      | Hybrid    | **85.54**	| **83.97**	| **83.78**	| **83.19** |
>
>
> B. On Left Atrium test set
>
> | Method | Annotation | 100%	| 50%	| 30%	| 10% |
> | :-----:| :----: | :----: | :----: | :----: | :----: |
> | KernelCut | Hybrid    | 77.54  | 76.72  | 73.64  | 67.27  |
> | Ours      | Hybrid    | **86.31**	| **86.25**	| **83.81**	| **83.41** |
>
>
> C. On Prostate val set
>
> | Method | Annotation | 100%	| 50%	| 30%	| 10% |
> | :-----:| :----: | :----: | :----: | :----: | :----: |
> | KernelCut | Hybrid    | 80.18 | 77.90 | 77.58 | 73.45 |
> | Ours      | Hybrid    | **86.01**	| **85.71**	| **85.56**	| **80.89** |
>
>
> D. BoxPrior on Prostate val set
>
> | Method | Annotation | 100%    |
> | :-----:| :----: | :----:  |
> | BoxPrior  | Hybrid (loose box only)   | 59.82   |
> | PCE  | Hybrid    | 73.15  |
> | BoxPrior+PCE  | Hybrid    | 72.56 |
> | Ours | Hybrid | **86.01** |
>
>
> On Hybrid, our method still outperforms KernelCut and BoxPrior, which verifies the advantages of our method by incorporating self-taught 3D shape prior.
>
> As for Boxprior, we conducted experiments on the 100% setting of prostate dataset. As suggested by the reviewer, we added a Partial Cross-Entropy (PCE) loss to Boxprior to train on Hybrid, as BoxPrior+PCE in Table D, and the result is much lower than ours on Hybrid. Moreover, to investigate the contribution of the PCE loss and the original BoxPrior loss, we conducted experiments of PCE loss only and BoxPrior loss only on Hybrid. As shown in Table D, with PCE loss only, the result is already higher than BoxPrior+PCE. Note that we tried extensive hyper-parameters tuning for BoxPrior+PCE, including $w$ parameter, weight of each loss term, and learning rate, based on their experiments on thick boxes in their code. To conclude, it is nontrivial to apply BoxPrior to Hybrid.
>
>
>
> > Comparing all methods on 3D box annotations.
>
> We respectively disagree that we should compare all methods on 3D box annotations. The reasons are as follows:
>
> - Firstly, it is nontrivial to propose a guideline for 3D box annotations and hard to estimate the cost of it. To our best knowledge, 3D boxes are much harder to annotate than 2D weak labels, as annotating a 3D box requires comparison among all slices, while annotating 2D weak labels mostly requires observation on single slices.
> - Secondly, 3D boxes will probably cause performance drop of methods developed on tight 2D boxes. 3D boxes usually lead to loose 2D boxes on most slices. However, looser boxes can lead to worse performance, as discussed in Table 2 of BoxPrior, where a margin of 10 pixels could cause a performance drop of 5% for their method. In contrast, we show statistics of length of all tight bounding boxes in Table 4 of our paper, where the standard deviation is usually much larger than 20 (if considering a margin of 10 on both sides).
> - Finally, our method and KernelCut are designed for weak labels that supply both foreground and background information, while 3D boxes only provide loose background information, which is even looser than the background provided by Hybrid or Scribble* discussed in our paper. We would anticipate worse performance of both our method and KernelCut on 3D boxes than on Hybrid or Scribble*, even if we apply these methods to 3D boxes in some way.
>
> In summary, drawing 3D boxes has no guarantee in reducing annotation cost, yet will probably hurt performance of these three methods, comparing to existing weak annotations. We agree that 3D boxes provide a unique pattern of label information, and it might be a valuable future direction for weakly supervised volumetric segmentation, with specific design for the information 3D boxes provided. But we would disagree that it is necessary to compare our method, KernelCut and Boxprior on 3D boxes in this paper.

---

> ### Author Response · Authors · 2021-03-18
> **Reply to reviewer 1 (Part 2)**
>
> > Training procedure
>
> We will add an algorithm for the training procedure in our revision:
>
> ```
> Training Procedure
> Input: Images, weak labels;
> Output: Network parameters of SSN and SDN;
> 1. Train Semantic Segmentation Network (SSN) with weak labels;
> 2. Train Shape Denoising Network (SDN)
>     a. compute confidence of all predicted masks by SSN on training split;
>     b. select the mask with the highest confidence as the self-taught shape representation;
>     c. augment the shape representation with designed noise and spatial transformation as input for SDN;
>     d. train SDN to reconstruct the clean shape;
> 3. Iterative learning
>     Repeat:
>       a. generate pseudo labels for current input, by combining outputs of SSN and SDN with uncertainty filtering;
>       b. update SSN with generated pseudo labels and weak labels;
>     Until reaching the maximum epoch
> ```
>
> > Training and inference time for SDN
>
> We train the SDN for 100 epochs on two TITAN Xp cards and the time cost is different for each dataset: 10h for prostate and 15-20h for trachea or left atrium. For inference of the SDN on a single TITAN Xp card, the cost is less than 0.02s for all datasets.
>
>
> **Detailed Comments:**
> > I personally think that the main contribution of the paper is the second-stage network, that can be trained to perform a "clean-up" of the predicted segmentation. I would therefore to present it as a pluggable (but still end-to-end trainable) module that can go after any base network (be it fully or weakly supervised).
>
> The main contribution of our paper is a novel pipeline for the task of weakly supervised volumetric segmentation, which includes two aspects: a self-taught learning method (our shape denoising network and its learning strategy) to capture the 3D shape prior of a target object class, that is further incorporated in an iterative learning framework, and a sparse annotation scheme (slice selection and hybrid label). We do not focus on developing a pluggable module for any base network, but aim to leverage shape prior information to improve segmentation learning with our SDN in weakly supervised settings.
>
>
>
> > Releasing code.
>
> We released our code on GitHub: https://github.com/Seolen/weak_seg_via_shape_model
>
> > The nuance of comparisons between 3D segmentation methods and their 2D counterparts.
>
> We thank reviewer 1 for pointing out the situations where 2D segmentation methods might have better performance than 3D methods, i.e., when dealing with 3D volumetric data with large inter-slices spacing or having insufficient 3D training data. We will clarify the comparisons and cite the related work in our revision.

---

> ### Author Response · Authors · 2021-03-18
> **Reply to reviewer 1 (Part 3)**
>
> **Questions To Address In The Rebuttal:**
> > The paper would benefit a lot from detailing more the self-taught shape denoising network, which is the main contribution of the paper. Could you develop that during the rebuttal, and highlight the differences with (Vincent et al., 2010) and (Sundermeyer et al., 2018) ?
>
> As addressed above, our main contribution is a novel pipeline for the task of weakly supervised volumetric segmentation and the self-taught shape denoising network is a part of it.
>
> The design of our SDN starts from the observation that some of the masks from initially trained SSN are in better shape than others, which can be selected based on the foreground confidence of each predicted mask. Then we can encode the shape prior and capture the underlying shape manifold with our SDN and noise augmentation design.
>
> We highlight the differences between our Shape Denoising Network (SDN) and Denoising Autoencoder (DAE) (Vincent et al., 2010) or Augmented Autoencoder (AAE) (Sundermeyer et al., 2018) as follows:
>
> - Our SDN is trained in a self-taught manner. We do not have ground truth training data or synthetic data to train our model, but utilize a self-taught shape representation extracted from weak labels with our SSN.
> - Our SDN captures an underlying shape manifold, while DAE captures an image manifold and AAE captures an orientation manifold.
> - We design noise augmentation based on the observation of typical errors in weakly supervised segmentation in a generic manner, which can apply to datasets with distinctive shape properties.
> - Our SDN aims for the reconstruted masks with clean shape, which encode shape prior and feed back to SSN for further improvement of the whole system, instead of a latent embedding.
>
> > How would your method generalize to multi-class settings ?
>
> In Section 2.1 Problem Setting and Model Overview of our paper, we explain that our method can be applied to multi-class settings by dealing with each class separately, i.e., treating each class as a binary segmentation problem and training an individual model for each foreground class.
>
> > How would your method behave if it was trained with only tight boxes ?
>
> As explained above, our method is designed for weak labels that supply both foreground and background information, which is more similar to the weak label setting of scribbles adopted by KernelCut. To apply our method to tight boxes only, we could first generate initial segmentation proposals for boxes resorting to some off-the-shelf tools and then treat these proposals as weak labels to apply our method. Specifically, we follow a similar strategy adopted by DeepCut and use GrabCut to generate our initial proposals. Due to the limitation of time and computation resources within a week, we only present results on Trachea in the following table. On Box, our method still achieves better performance than BoxPrior.
>
>
> | Method | Annotation | 100%	| 50%	| 30%	| 10% |
> | :-----:| :----: | :----: | :----: | :----: | :----: |
> | BoxPrior	| Box   | 79.82	| 48.78	| --	| --    |
> | Ours	| Box	    | 83.67	| 81.64	| 80.36	| 80.16 |
> | Ours	| Hybrid	| **85.54**	| **83.97**	| **83.78**	| **83.19** |

---

### Official Review · AnonReviewer2 · 2021-03-09

**Confidence:** 4
**Preliminary Rating:** 3
**Recommendation:** Oral, Poster

**Summary:**

The paper addresses an important problem of annotation cost in deep learning. The author proposes a weakly supervised method for medical image segmentation. The authors claim that earlier methods haven't fully exploited the volumetric nature of medical images. Thereby, the authors propose a weakly-supervised segmentation strategy through shape priors. Unlike other works, the authors have proposed a self-taught shape representation through weak labels and use that to refine the segmentation.The authors have proposed an iterative strategy to train the segmentation module and the shape module. They have also proposed a hybrid label design without increasing the annotation cost. The proposed methods have shown promising results

**Strengths:**

* The paper is well written and easy to follow.
* The paper addresses an important challenge in medical image analysis.
* The paper proposes as a set of intuitive steps to provide segmentation with weak labels. The proposed method is simple, but it looks efficient.
* The strategy of self-training instead of having a synthetic model is an interesting direction.
* The proposed method has been evaluated with three datasets and an ablative study on different annotation strategies has been conducted


**Weaknesses:**

* The authors have shrewdly proposed a pipeline with various existing methods. But, the chosen methods are already existing ones. Hence, the novelty is not very substantial.
* In my opinion, the self-taught mask is the important part of this work, more focus could have been given to that.


**Deanonymize Review:**

no

**Justification Of The Preliminary Rating:**

The author propose a method to address an important problem in medical image analysis. The proposed pipeline provides promising results. The self-training of shape model and the annotation strategy are interesting directions.

**Paper Type:**

methodological development

**Special Issue:**

no

---

> ### Author Response · Authors · 2021-03-18
> **Reply to reviewer 2**
>
> We thank reviewer 2 for the positive comments and the appreciation for our self-training strategy. The main concerns are about the novelty of our method. Please find our detailed explanations as below.
>
> **Weaknesses:**
> > The authors have shrewdly proposed a pipeline with various existing methods. But, the chosen methods are already existing ones. Hence, the novelty is not very substantial.
>
> We propose a novel pipeline for the task of weakly supervised volumetric segmentation, which achieves superior performance over other weakly supervised methods under the same labeling cost. The novelty of our method includes two aspects: a self-taught learning method (our shape denoising network and its learning strategy) to capture the 3D shape prior of a target object class, which is further incorporated in an iterative learning framework, and a sparse annotation scheme (slice selection and hybrid label). Our self-taught learning strategy is able to extract a shape representation from weak labels based on initially trained Semantic Segmentation Network (SSN), which we believe has not been explored before.
>
>
>
>
> > In my opinion, the self-taught mask is the important part of this work, more focus could have been given to that.
>
> We would like to further discuss the motivation, observation and intuition of our self-taught mask.
>
> - The key motivation of the self-taught learning method is to leverage predicted masks in good quality to refine predictions with lower accuracy.
> - Empirically, we found the SSN trained on our weak labels is able to predict masks with better shape quality for some training instances, producing cleaner and more complete object shape.
> Additionally, we found that the shape quality is closely related to the foreground confidence of SSN predictions, and the overall shape variations within a dataset can be partially captured by augmenting one shape with spatial transformation.
> - Therefore we take the mask with the highest quality and design a generic noise augmentation to train our SDN, which captures an underlying shape manifold and performs denoising on input masks.
>
> As the self-taught mask is one of our main contributions, we spent about one page to discuss the model design and model learning for our SDN. Due to the page limit, we presented further analysis on this module in Appendix F.3. We appreciate the suggestion and will include more discussion in our revision.

---

### Official Review · AnonReviewer4 · 2021-03-09

**Confidence:** 4
**Preliminary Rating:** 3
**Final Rating:** 3

**Summary:**

This paper presents a weakly supervised volumetric segmentation method based on self-taught shape denoising. The proposed model is composed of two modules, a semantic segmentation network (SSN) and a shape denoising network (SDN), trained in an iterative manner. The SSN is trained with weak annotations and pseudo-labels generated from the SDN. On the other hand, the SDN uses an autoencoder to learn a shape prior from the confident segmentation masks of the SSN. Evaluation on three benchmarks with distinctive shape properties shows that the proposed method outperforms other recently proposed approaches for weakly-supervised segmentation.

**Strengths:**

* The proposed weak annotation scheme with hybrid label design is interesting. It requires only 4 point for each slice and improves performance without increasing the overall annotation cost. Experiments show the benefits of using this hybrid annotation scheme.

* Although several iterative pseudo-label schemes have been proposed for weakly-supervised segmentation, the proposed method differs in its use of an autoencoder trained with noise augmentation.

* Experiments include several ablation studies evaluating the contribution of individual components of the model.


**Weaknesses:**

* Despite some architectural differences, the technical contributions of this work are somewhat limited. Iterative pseudo-label generation is at the core of many weakly-supervised segmentation methods, such as DeepCut. Using autoencoders to model segmentation shape priors is also a well-known strategy, e.g. used in Oktay et al.:  Anatomically constrained neural networks (ACNNs): application to cardiac image enhancement and segmentation.  IEEE transactions on medical imaging, 2017. As I understand, the main contribution is the self-taught technique which uses the confident predictions of the segmentation network and noisy augmentation to train the autoencoder (SDN). However, this is done in an adhoc manner (selection of a single reference mask and no iterative update of the SDN), and it is unclear how this initial segmentation can boost training.

* The presentation of the method is quite dense and sometimes hard to follow. Important details about the annotation and training strategy are left in the Appendix.

* Lack of a proper discussion that highlights the limitations of the proposed method and potential improvements.

**Deanonymize Review:**

no

**Detailed Comments:**

* The overview of the proposed method can be improved. In particular, it would be useful to include the different training steps in the diagram.

* As shown in Figure 2, there are three different annotations. In the experiments, the author give the performance of using the scribble and the proposed annotations. Is it possible to conduct a similar experiment using only bounding boxes?

* It is mentioned that an auto-weighting strategy is used to balance foreground and background regions. More details are needed to understand this strategy.

* Based on the weights, the importance of weakly supervised loss term is much smaller (1:100) than the second term in SSN training. What would happen if you remove the weak-supervision term?

* p1 "...they ignore the continuity of volumetric data in 3D space and are unable to exploit label correlation between consecutive 2D slices, which often leads to additional cost from.." This problem has been addressed in existing literature, for example see:

Peng, J., Kervadec, H., Dolz, J., Ayed, I. B., Pedersoli, M., & Desrosiers, C. (2020). Discretely-constrained deep network for weakly supervised segmentation. Neural Networks, 130, 297-308.

* In the self-taught shape prior, why do you use a single shape representation? Is it enough to capture the variability of shapes, even with augmentation?

**Final Rating Justification:**

I read through the responses and other reviewers' comments. The authors have answered my main comments and comments. I encourage authors to add in their final manuscript the additional discussion, algorithm and experiments suggested by reviewers. If possible, the manuscript should further explain the “top 1,3 or 5 shapes” and the corresponding experimental results.

**Justification Of The Preliminary Rating:**

Overall, this paper presents a solution for a general problem (weak supervised volumetric medical segmentation) . The experimental results show some improvement compared to other approaches. However, there are some issues regarding the novelty and motivation of the proposed method.

**Paper Type:**

methodological development

**Questions To Address In The Rebuttal:**

* Emphasize the novelty of the proposed method and explain how using a single shape representation obtained from an initial segmentation can improve overall accuracy. Improve the overall presentation of the method.

**Special Issue:**

no

---

> ### Author Response · Authors · 2021-03-18
> **Reply to reviewer 4 (Part 1)**
>
> We thank reviewer 4 for the positive comments on our weak annotation scheme, self-taught shape denoising method and experimental results. The main concerns are about the novelty of our method and some improvement for presentation. Please find our detailed explanations as below.
>
> **Weaknesses:**
>
> > Despite some architectural differences, the technical contributions of this work are somewhat limited.
> > Iterative pseudo-label generation is at the core of many weakly-supervised segmentation methods, such as DeepCut.
> > Using autoencoders to model segmentation shape priors is also a well-known strategy, e.g. used in Oktay et al.: Anatomically constrained neural networks (ACNNs): application to cardiac image enhancement and segmentation. IEEE transactions on medical imaging, 2017. As I understand, the main contribution is the self-taught technique which uses the confident predictions of the segmentation network and noisy augmentation to train the autoencoder (SDN). However, this is done in an adhoc manner (selection of a single reference mask and no iterative update of the SDN), and it is unclear how this initial segmentation can boost training.
>
> The main contribution of our paper is a novel pipeline for the task of weakly supervised volumetric segmentation, capable of achieving SOTA performance comparing to other weakly supervised methods under the same labeling cost. The novelty of our pipeline includes two aspects: a self-taught learning method (our shape denoising network and its learning strategy) to capture the 3D shape prior of a target object class, which is further incorporated in an iterative learning framework, and a sparse annotation scheme (slice selection and hybrid label).
>
> Iterative pseudo-label generation is indeed at the core of many weakly-supervised segmentation methods, while we incorporate a self-taught shape prior with an uncertainty filtering mechanism to improve the quality of generated pseudo labels, which is unique and unexplored.
>
> ACNNs does investigate modeling shape prior with an autoencoder for fully supervised segmentation and image super resolution. While our work is substantially different from their method in several aspects:
>
> - Firstly, we develop a self-taught learning method. Given only weak labels, our method is able to extract a self-taught shape representation by exploiting the confidence of segmentation output. With this shape representation, we apply spatial transformation to cover a wide range of the shape variations and form the manifold of the true shape data. Note that this self-taught strategy significantly reduces labeling cost comparing to the full supervision adopted by ACNNs.
> - Secondly, we design a shape denoising network to capture the underlying manifold of the true shape. In contrast to adding regularization constraints in encoding space, our method aims to explicitly perform denoising and to recover the clean shape.
>
> We thank the reviewer for pointing out this and will add comparison to ACNNs in our revision.
>
> Additionally, our noise augmentation is based on the observation of typical errors in weakly supervised segmentation and designed in a generic manner, which can apply to datasets with distinctive shape properties. This is well-motivated by the task of weakly supervised volumetric segmentation for medical images.
>
>
> Given this initial segmentation with high quality in shape, we design a noise augmentation strategy and train our SDN to capture the underlying shape manifold and to perform denoising. After the training of SDN, the initially predicted masks of SSN with worse shape quality than the selected mask can be improved by the trained SDN. Thus, we can generate pseudo labels with both outputs of SSN and SDN to improve supervision and to encode shape prior for segmentation learning.
>
>
>
> > The presentation of the method is quite dense and sometimes hard to follow. Important details about the annotation and training strategy are left in the Appendix.
>
> We present some of the details of the method in Appendix mainly due to the page limit. We will improve this in our revision.
>
> > Lack of a proper discussion that highlights the limitations of the proposed method and potential improvements.
>
> We discuss the failure case and future work in Appendix F.4. Our method explored high-level shape prior of objects in volumetric images, while low-level features such as contour information are unexplored in our work. A potential improvement would be to combine
> high-level shape prior and these low-level image features for better segmentation results.

---

> ### Author Response · Authors · 2021-03-18
> **Reply to reviewer 4 (Part 2)**
>
> **Detailed Comments:**
>
> > The overview of the proposed method can be improved. In particular, it would be useful to include the different training steps in the diagram.
>
> We thank the reviewer for the suggestion and will improve the overview in Figure 1 by adding the different training steps and corresponding descriptions in our revision.
>
> > Is it possible to conduct a similar experiment using only bounding boxes?
>
> Our method is originally designed for weak labels that supply both foreground and background information, which is more similar to the weak label setting of scribbles adopted by KernelCut. To apply our method to bounding boxes only, we could first generate initial segmentation proposals for boxes resorting to some off-the-shelf tools and then treat these proposals as weak labels to apply our method. Specifically, we follow a similar strategy adopted by DeepCut and use GrabCut to generate our initial proposals. Due to the limitation of time and computation resources within a week, we only present results on Trachea in the following table. On Box, our method still achieves better performance than BoxPrior.
>
>
> | Method | Annotation | 100%	| 50%	| 30%	| 10% |
> | :-----:| :----: | :----: | :----: | :----: | :----: |
> | BoxPrior	| Box   | 79.82	| 48.78	| --	| --    |
> | Ours	| Box	    | 83.67	| 81.64	| 80.36	| 80.16 |
> | Ours	| Hybrid	| **85.54**	| **83.97**	| **83.78**	| **83.19** |
>
>
>
> > It is mentioned that an auto-weighting strategy is used to balance foreground and background regions. More details are needed to understand this strategy.
>
> As the weak labels provide background labels on all background slices and regions outside loose boxes, while only provide foreground labels on scribbles, the imbalance between background and foreground is quite severe and varies for different datasets. Therefore, we take an auto-weighting strategy for weighted cross-entropy loss, to automatically adapt to different imbalance situations and to avoid tedious hyper-parameter tuning. Specifically, for each minibatch, assume the number of labeled background pixels is $N_b$ and the number of labeled foreground pixels is $N_f$. The loss would be $(1 / N_b \sum^{N_b}_i l_i + 1 / N_f \sum^{N_f}_j l_j) / 2$, where $l_i$ denotes the cross-entropy loss of pixel i. Empirically we found this strategy works better than a constant weight.
>
>
>
> > Based on the weights, the importance of weakly supervised loss term is much smaller (1:100) than the second term in SSN training. What would happen if you remove the weak-supervision term?
>
> Please check Appendix F.2. and Table 7, where we show that removing the weak-supervision term will cause a performance drop of 1%-2% in dice.
>
>
> > p1 "...they ignore the continuity of volumetric data in 3D space and are unable to exploit label correlation between consecutive 2D slices, which often leads to additional cost from.." This problem has been addressed in existing literature.
>
> We thank the reviewer for pointing out this. In our revision, we will modify the description and cite the related works, e.g., [1].
>
> [1] Peng, J., Kervadec, H., Dolz, J., Ayed, I. B., Pedersoli, M., & Desrosiers, C. (2020). Discretely-constrained deep network for weakly supervised segmentation. Neural Networks, 130, 297-308.
>
>
> > In the self-taught shape prior, why do you use a single shape representation? Is it enough to capture the variability of shapes, even with augmentation?
>
> From the observation of initially trained SSN, we found that organs within a dataset typically have very similar shapes. The main variations can be captured by some mild spatial transformations, including translation, rotation, and scale. Therefore, we augment a single selected shape with random spatial transformation to form the shape distribution. Empirically we found that using a single shape with augmentation is sufficient for this task. We conducted experiments where we selected top 1, 3, or 5 shapes with the highest confidence to train the SDN, and the results of these three settings are comparable. For volumetric segmentation with large variety in shape, our method can potentially extend to multi-shape learning, e.g., resorting to standard clustering methods to obtain several templates, training an independent autoencoder for each shape, and obtain final results by voting based on similarity.
>
>
>
> **Questions To Address In The Rebuttal:**
>
> > Emphasize the novelty of the proposed method and explain how using a single shape representation obtained from an initial segmentation can improve overall accuracy. Improve the overall presentation of the method.
>
> Please find the detailed explanations for these issues as above.

---

### Meta-Review · Area_Chairs · 2021-03-29

**Recommendation:** Accept (Poster)

**Metareview:**

The authors present a new method for performing weakly supervised segmentation using shape models. The authors build on several ideas (autoencoders for models, etc) to put together a pipeline that gets the job done.

Most reviewers had several concerns and noted many missing literature aspects. While the rebuttal helped, I am worried that the authors in some cases answered by emphasizing the difference to the mentioned paper, such as Oktay et al. I want to emphasize that this paper, and other anatomical prior papers especially ones that use VAEs, should have been cited from the beginning even if they exhibit differences to the current method -- the idea is to explain which method are similar or you build on and give appropriate credit.

Having said this, I commend the authors on their thorough rebuttals which addressed a lot of the concerns and clarified a lot of the problems of the reviewers. All reviewers in the end recommend a weak accept, but emphasize several issues and the accepts are conditional on the significant improvements made during the rebuttal process. I therefore recommend conditional accept, conditioned on the authors incorporating all the promised changes and ensuring that all the feedback is taken into account. It is especially important here as the reviewers were thorough and the rebuttals were part of the final decision.

**Paper Type:**

methodological development

---

### Decision · Program_Chairs · 2021-03-31

Accept